# Intelligibility improves perception of timing changes in speech

**Benedikt Zoefel**[1,2,3]*, **Rebecca A. Gilbert**[1], **Matthew H. Davis**[1]

**1** MRC Cognition and Brain Sciences Unit, University of Cambridge, Cambridge, United Kingdom, **2** Centre National de la Recherche Scientifique (CNRS), Centre de Recherche Cerveau et Cognition (CerCo), Toulouse, France, **3** Université de Toulouse III Paul Sabatier, Toulouse, France

* benedikt.zoefel@cnrs.fr

**Data Availability Statement:** Data are available on the Open Science Framework repository (https://osf.io/p2ch8/).

**Funding:** This work was supported by the European Union's Horizon 2020 research and innovation programme under the Marie

## Abstract

Auditory rhythms are ubiquitous in music, speech, and other everyday sounds. Yet, it is unclear how perceived rhythms arise from the repeating structure of sounds. For speech, it is unclear whether rhythm is solely derived from acoustic properties (e.g., rapid amplitude changes), or if it is also influenced by the linguistic units (syllables, words, etc.) that listeners extract from intelligible speech. Here, we present three experiments in which participants were asked to detect an irregularity in rhythmically spoken speech sequences. In each experiment, we reduce the number of possible stimulus properties that differ between intelligible and unintelligible speech sounds and show that these acoustically-matched intelligibility conditions nonetheless lead to differences in rhythm perception. In Experiment 1, we replicate a previous study showing that rhythm perception is improved for intelligible (16-channel vocoded) as compared to unintelligible (1-channel vocoded) speech–despite near-identical broadband amplitude modulations. In Experiment 2, we use spectrally-rotated 16-channel speech to show the effect of intelligibility cannot be explained by differences in spectral complexity. In Experiment 3, we compare rhythm perception for sine-wave speech signals when they are heard as non-speech (for naïve listeners), and subsequent to training, when identical sounds are perceived as speech. In all cases, detection of rhythmic regularity is enhanced when participants perceive the stimulus as speech compared to when they do not. Together, these findings demonstrate that intelligibility enhances the perception of timing changes in speech, which is hence linked to processes that extract abstract linguistic units from sound.

## Introduction

Time is a highly important dimension for the auditory system: two short auditory stimuli (clicks) can be distinguished even if they are only presented 2 ms apart–in contrast to vision, where this "fusion threshold" is ~ 20 ms [1]. Rhythm is particularly crucial when processing a continuous stream of rapidly-fluctuating auditory information, as the timing of events in a rhythmic stream can, by definition, be more easily predicted. Indeed, rhythm is ubiquitous in the auditory environment, not only in human-produced sounds such as speech or music, but also in animal sounds, mechanical sounds, or in other environmental noises. However,

Sklodowska-Curie grant agreement number 743482 (to BZ), the Medical Research Council UK (grant number MC_UU_00005/5) (to MHD), and the Centre National de la Recherche Scientifique (to BZ). The funders had no role in study design, data collection and analysis, decision to publish, or preparation of the manuscript.

**Competing interests:** The authors have declared that no competing interests exist.

whereas defining auditory rhythms for musical or mechanical sounds is relatively straight-forward, establishing the acoustic or linguistic cause(s) of speech rhythm has remained challenging.

Although naturally-produced human speech is not perfectly acoustically regular, it does evoke a quasi-rhythmic perceptual experience [2, 3]. Indeed, "speech rhythm" has been a subject of intense investigation for decades [4–6]. An important phenomenon in this field of research is the "perceptual centre", or "p-centre", the perceptual moment of occurrence of a speech sound [7, 8]. When participants are asked to speak rhythmically, the p-centre is the part of the word or syllable that is aligned with each (e.g., metronome) beat. According to this definition, a speech stimulus is perceived as rhythmic if its p-centres are equally spaced in time. Despite this straightforward definition, it has been surprisingly difficult to reveal an acoustic correlate of the p-centre: Speech, constructed to be perceptually isochronous, is not acoustically isochronous [7–9]. Although more complex models exist to explain p-centres based on acoustic stimulus properties [10], this has produced a vivid debate [9, 11], and been complicated by the fact that various models do not seem to converge on a specific acoustic pattern linked to the p-centre [8–10]. Indeed, it has been proposed that the acoustic speech signal is only a carrier of more abstract linguistic information, and participants produce rhythmic speech by aligning that information–rather than its carrier–to the beat [11]. Accordingly, linguistic information seems to be an important determinant of the p-centre and speech rhythm perception.

There is no doubt that low-level acoustic properties, such as amplitude modulations, also play an important role for auditory and speech perception. Slow fluctuations (~2–6 Hz) in sound amplitude are prominent in speech sounds and are related [12], but not identical [13, 14], to the syllabic and prosodic rate. Human sensitivity to temporal (amplitude or frequency) modulations of acoustic input peaks at that frequency range [15], and the human auditory system seems specialized in processing such amplitude modulations [16, 17]. Removing slow amplitude modulations (e.g., using low-pass filters) strongly reduces speech comprehension [18]. Speech that is time-compressed (e.g., average syllable rate 9 Hz), and therefore unintelligible, can be made intelligible by inserting silent periods so that the overall rhythm is closer to that of typical speech (e.g., average syllable rate 6 Hz [19, 20]).

Interestingly, however, slow amplitude modulations are not sufficient to produce reliable speech perception. Studies using noise-vocoding, a technique introduced in 1995 by Shannon and colleagues [21], have been crucial to reach this conclusion. During noise-vocoding, the speech signal is divided into a certain number of frequency bands ("channels"). Amplitude modulations (i.e. envelopes) are then extracted for each of those channels and applied to white noise, filtered into the corresponding frequency ranges. These noises are re-combined to yield noise-vocoded speech stimuli. The amount of spectral detail of these stimuli increases with the number of channels, but without affecting their broadband amplitude envelope, i.e. the slow amplitude fluctuations described above. The number of channels also determines whether noise-vocoded stimuli are intelligible: 16-channel noise-vocoded speech is clearly intelligible, even for naïve listeners, whereas 1-channel noise-vocoded speech sounds like noise to naïve listeners and is completely unintelligible when presented in isolation. 1-channel vocoded speech yields <5% word report accuracy when presented alone, but ~30% when combined with visual cues like lip movements [22], suggesting that noise-vocoded speech can be perceived as speech-like, even when unintelligible. Importantly, however, these manipulations reveal changes in speech intelligibility that are independent of amplitude modulations, since these are very similar for 1-channel and 16-channel speech [23]. Other brain imaging studies have used noise-vocoded speech to demonstrate that amplitude modulations alone are not sufficient to activate brain regions identified as specifically processing human speech [24, 25],

again suggesting that additional stimulus properties are necessary to drive the perception of speech and its rhythm.

## Current study

As detailed above, literature on the perceptual centre suggests that perception of auditory rhythms, and of speech rhythm in particular, is not established exclusively by certain acoustic signals, and instead hints at an important role of linguistic properties [9, 11]. However, there have been few direct investigations of whether linguistic properties improve the perception of auditory rhythms, above and beyond that evoked by acoustic properties. In this study, we take an approach that differs from previous work on speech rhythm perception. We explicitly refrain from testing or manipulating speech rhythm perception per se. Rather, we test how certain acoustic and perceptual manipulations that change the intelligibility of perceptually rhythmic speech also affect our ability to detect irregularities in that rhythm. Our rhythmic speech is not as complex as connected speech, but nonetheless our stimuli are derived from naturally-produced speech and ecological: when counting, or reciting the alphabet, speakers will produce similar sequences that listeners perceive as rhythmically-regular monosyllables. This detail is important as our speech stimulus is already perceptually rhythmic–as explained below, the well-defined (but, admittedly, less complex) rhythm allows a straightforward quantification of how these manipulations affect rhythm perception.While noise-vocoding has been used to successfully manipulate speech intelligibility independently of slow amplitude fluctuations [21, 23, 24], this technique has not been used much in research on speech rhythm perception. In one previous study [26], we designed perceptually-isochronous noise-vocoded speech sequences (one-syllable words, spoken based on a metronome beat) and asked participants to detect a rhythmic irregularity. We found that it was easier for participants to detect such a violation from isochrony when the speech was intelligible (16-channel noise-vocoded speech, henceforth shortened to 16-channel speech) than when it was not (1-channel noise-vocoded speech, 1-channel speech). This finding was initially surprising, given that slow amplitude modulations are extremely similar in intelligible and unintelligible conditions and equally preserve those present in clear speech. Indeed, since all frequency bands are co-modulated in 1-channel speech (see Experiment 1, Materials) we might have expected that this unintelligible condition could evoke a *stronger* percept of rhythm than 16-channel speech, in which co-modulation between frequency bands is reduced (see ref [23], for further discussion of amplitude co-modulation for vocoded speech).

This result from our previous study [26] represents tentative evidence that linguistic properties contribute to auditory rhythm perception. In the current study, we followed up on this result. As in our previous study, we constructed sequences of five one-syllable words, spoken based on a metronome beat. We defined the p-centre of these words as that part of the word that the speaker (author MHD) aligned to the metronome beat. We assume that this led to p-centres in our five-word sequences being isochronous, and therefore rhythmic in perceptual terms. We then introduced irregularities in these rhythmic sequences by shifting one of the words towards another, and asked participants to decide whether they detected a rhythmic irregularity ("irregular trial") or not ("regular trial"). Note that our definition of "irregular" is entirely based on this temporal shift rather than p-centre positions of individual words (see General Discussion). Moreover, although some imperfection during the metronome recording cannot be ruled out, imperfectly spoken stimuli (i.e. not perfectly aligned to the isochronous metronome beat) would only reduce detection of irregularities overall. This would not bias outcomes in specific experimental conditions, since the same set of stimuli were used in all conditions.

In this study, we carried out three experiments to further investigate how the linguistic properties of rhythmic speech affect the ability to detect violations in the stimulus rhythm, and to discard alternative explanations for the observed effects. In each of these experiments, described below, we manipulated certain properties of the rhythmic sequences. We then tested how performance in our irregularity detection (i.e. rhythm perception) task was affected by these manipulations and their relationship with the intelligibility of the speech sequences. In Experiment 1, we aimed to replicate our original finding of improved rhythm perception during intelligible (16-channel) speech [26], using a higher proportion of irregular trials and in a forced-choice paradigm. In Experiment 2, we tested whether differences in spectral complexity can explain the observed differences between 16-channel and 1-channel speech. In Experiment 3, we compared rhythm perception between one experimental group trained to perceive a (sine-wave speech) stimulus as speech and another group which was not trained. This allowed us to contrast rhythm perception between stimuli which were acoustically identical but differed in their intelligibility. Results from all experiments point towards an important role of linguistic properties for auditory rhythm perception.

## Experiment 1: Better rhythm perception for 16-channel versus 1-channel vocoded speech

In our previous study [26], the irregularity detection task was included as an incidental part of an fMRI study; it was intended to keep participants alert and hence only contained a small number of irregular trials (~14%) on which participants made a button press response (they did not press a button during regular trials). For all of the experiments described here, we increased this number to 50% and required a forced choice (irregular vs regular) response on all trials. This allowed us to more reliably estimate participants' ability to correctly detect irregularities ("hits") as well as the tendency to incorrectly classify regular sequences as irregular ("false alarms"). Together, these two estimates can be used to calculate d-prime, a measure of perceptual sensitivity, and independently we can measure criterion or response bias; participant's overall tendency to respond that sequences are, or are not regular (see Experiment 1, Statistical Analysis).

We acquired the data for experiments 2 and 3 online, which made for faster and more efficient data collection without loss of data quality [27]. However, given the difference in methods here compared to the initial study [26], we first tested whether we could replicate the intelligibility effect in a laboratory setting. This was the purpose of Experiment 1. We expected to find higher sensitivity (d-prime) to detect rhythmic irregularities during intelligible (16-channel) than during unintelligible (1-channel) speech.

### Methods

**Participants.**   Six participants (3 female; M 28.2 years, SD 4.1 years, range 23–32 years) completed the experiment after giving written informed consent under a process approved by the Cambridge Psychology Research Ethics Committee (application number PRE.2015.132). All participants were fluent speakers of English.

The number of participants was kept relatively low, because Experiment 1 was designed to pilot the change in certain experimental parameters (forced choice paradigm with 50% target trials) with respect to previous work [26], prior to data collection online (Experiments 2 and 3).

**Materials.**   We used the same rhythmic speech sequences as described in detail in ref [26]. The original speech (Fig 1A) consisted of five one-syllable words, recorded in time with a 2-Hz metronome beat by a male speaker of Standard Southern British English (author MHD). The

speech was then time-compressed such that the syllable rate was 3.125 Hz (reducing the original 500 ms intervals to 320 ms; 64% of the original duration), using the pitch-synchronous overlap and add (PSOLA) algorithm implemented in the Praat software package (version 6.12). Individual words were extracted and combined into rhythmic sentences of five words, in the following order: "pick" <number> <colour> <animal> "up", where <number> could be any number between one and nine (excluding the bisyllabic word "seven"); <colour> could be any of: "black", "green", "blue", gold", "red", "grey", "pink", "white"; and <animal> could be any of: "bat", "frog", "cow", "dog", "fish", "cat", "sheep", "pig". Each five-word sequence was 1.6 seconds long.

In 50% of the trials, one of the three middle words in the stimulus was shifted in time (either forwards or backwards, with equal probability). The size of the shift was ±68 ms for all participants and conditions. This was the same manipulation used in our initial study [26], in which this shift size resulted in moderate detection performance (d-prime of ~2, see below). This moderate difficulty level should be sufficient to detect between-condition differences in sensitivity to irregularities in speech rhythm.

We tested participants' ability to detect irregular stimulus rhythms in two experimental conditions that differed in the intelligibility of the rhythmic speech sequences. We manipulated the five-word sequences using noise-vocoding [21]. The speech signal was first filtered into 1 and 16 logarithmically-spaced frequency bands (channels) between 70 and 5000 Hz, for the 1-channel and 16-channel conditions, respectively. The amplitude envelopes were extracted in each band by half-wave rectification and low-pass filtering below 30 Hz such that slow amplitude modulations assumed to be relevant for speech rhythm perception are preserved. These amplitude envelopes were then applied to white noise, filtered into the corresponding frequency ranges, and the output signals were re-combined to yield noise-vocoded speech stimuli. As explained above, the intelligibility of these stimuli increases with the number of channels used, without affecting their broadband amplitude envelopes. In this experiment, as in our initial study [26], we contrasted intelligible 16-channel speech (Fig 1C), which we expected to be highly intelligible, even for naïve listeners, with noise-like, unintelligible 1-channel speech (Fig 1B), which we expected to be almost entirely unintelligible (see refs [23, 28], for evidence of floor and ceiling word report scores for sentences and single words processed in similar ways).

**Procedure.** Participants used Sennheiser HD 202 headphones to listen to the rhythmic sequences, delivered by a standard sound card at an individually comfortable level. They completed 200 trials for each of the 16-channel and 1-channel conditions. Condition (16- vs 1-channel) as well as the presence of an irregularity, shifted syllable number (second, third or fourth), and shift direction (forwards or backwards) was determined pseudo-randomly for each trial. Note that, although irregularities were introduced pseudo-randomly, some might have been more salient than others: For instance, for irregular second syllables, only the occurrence of the third syllable creates the percept of irregularity. These differences, however, were present for all conditions and cannot affect comparisons between them. After each trial, participants indicated whether or not they detected an irregularity in the stimulus rhythm, using two different keys on a computer keyboard. The experiment continued with the next trial after each button press. The task was explained to the participants using four example stimuli (one for each combination of the two conditions and regular vs irregular trials). No further training was provided and no feedback was given. The experiment was not divided into blocks, but participants were able to take breaks between trials.

**Statistical analysis.** The participants' sensitivity to detect an irregularity in the stimulus rhythm was quantified using d-prime (*d'*), computed as the standardized difference between

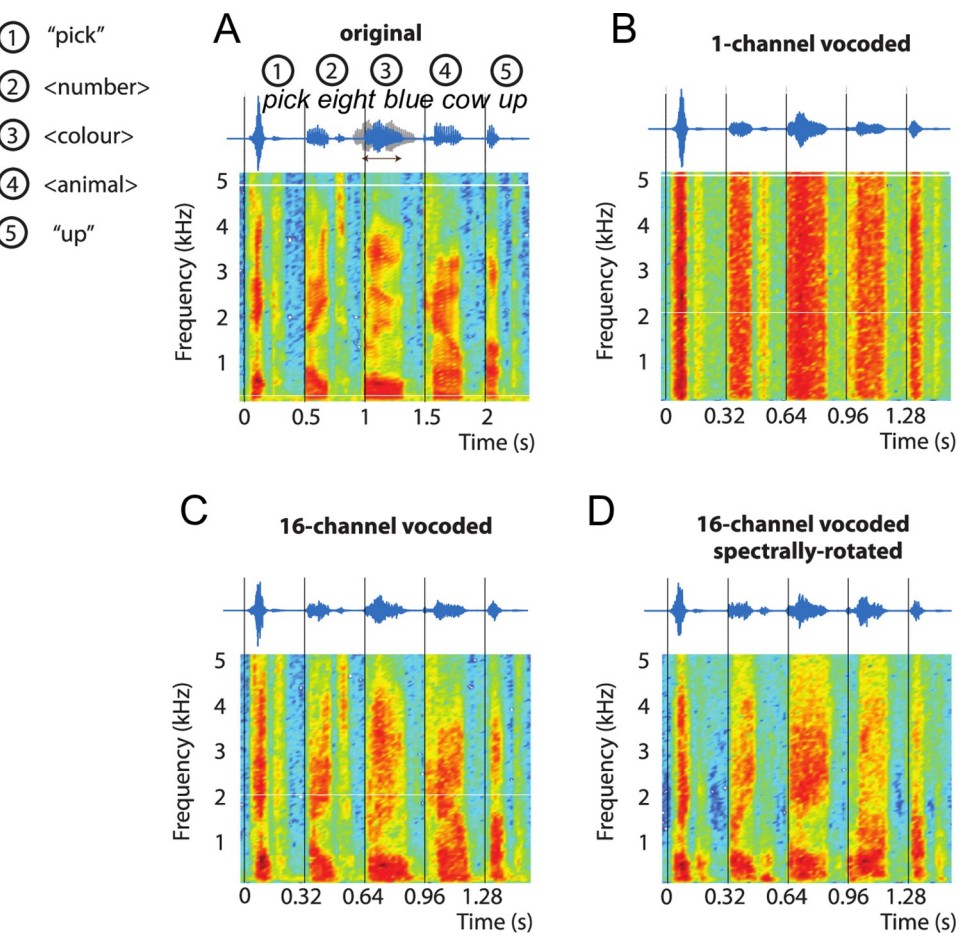

**Fig 1. Stimuli used in Experiments 1 and 2.** The original speech (A) consisted of five one-syllable words, spoken at 2 Hz based on a metronome beat. This led to perfectly rhythmic speech, based on perceptual terms, with p-centres aligned with the metronome beats (vertical lines). First and last words of the five-word sequence were always "pick" and "up", respectively. The other three words were a number, colour, and animal, with eight possible options for each (see Experiment 1, Materials)–"pick eight blue cow up" for all panels in this figure. Original speech was time-compressed to 3.125 Hz for all experiments. In 50% of the trials, the second, third, or fourth word was shifted by ±68 ms to create an irregularity in the stimulus rhythm which participants were asked to detect (example waveforms shown in grey marked with an arrow). For Experiments 1 and 2, stimuli shown in A were modified using noise-vocoding, yielding unintelligible 1-channel vocoded speech (B) and intelligible 16-channel vocoded speech (C) (Experiment 1, Materials). For Experiment 2 only, amplitude envelopes extracted for the construction of noise-vocoded speech were rotated before being applied to noise, yielding unintelligible, but spectrally complex, 16-channel rotated speech (D) (Experiment 2, Materials).

hit probability and false alarm probability:

$$d' = z(p_{hit}) - z(p_{false\ alarm})$$

where

$$p_{hit} = \frac{N_{hits}}{N_{targets}}$$

$$p_{false\ alarm} = \frac{N_{false\ alarms}}{N_{trials} - N_{targets}}$$

and $N_{hits}$, $N_{false\ alarms}$, $N_{targets}$ and $N_{trials}$ represent the number of hits (correctly identified irregularities), the number of false alarms (regular trial incorrectly identified as irregular), the total number of irregularities, and the total number of trials, respectively, in a given condition. Note that given the $z$-transformation, $d'$ is not defined for hit or false alarm rates of 0 or 1. In this case, $\frac{0.5}{N_{targets}}$ was added to (for rates of 0), or subtracted from (for rates of 1), both $N_{hits}$ and $N_{false\ alarms}$, respectively [29].

Independently of participants' ability to identify irregularities (i.e. $d'$), they might also differ in their bias towards giving a specific response (i.e. "regular" or "irregular"). This decision bias was estimated as criterion $c$, commonly used in Signal Detection Theory [30]:

$$c = -0.5 * (z(p_{hit}) + z(p_{false\ alarm}))$$

A negative $c$ reflects a participant's "liberal" strategy, i.e. they respond with "irregular" more often than appropriate (resulting in a higher than optimal false alarm rate but a high hit rate). A positive $c$ reflects a more "conservative" strategy, i.e. they respond with "regular" more often (resulting in a lower than optimal hit rate but relatively few false alarms).

We varied the position (second, third or fourth word) and direction (shifted forwards vs backwards) of the irregularities in the five-word sequence (see Materials). However, since $p_{false\ alarm}$ is determined from regular trials, for which neither position nor direction is defined, it is not possible to determine $p_{false\ alarm}$ (and therefore neither $d'$ nor $c$) separately for different target positions or shift directions. We therefore pooled trials across these variables before comparing performance measures ($d'$ and $c$) between intelligibility conditions, using repeated-measures $t$-tests and ANOVAs. Because all stimulus items were equally likely to appear in all experimental conditions, between-item variance will also contribute to these ANOVAs and hence conventional by-participant analyses will appropriately control Type II error rates [31]. MATLAB 2014a (The MathWorks, Inc.) was used for stimulus presentation and for all the statistical analyses described.

## Results

Results for individual participants and for the group are shown in Fig 2. Perceptual sensitivity to the difference between rhythmically regular and irregular trials (d-prime) was significantly higher for intelligible, 16-channel speech than for unintelligible, 1-channel speech ($t(5) = 2.75$, $p = 0.04$; effect size, Cohen's d = 1.12). The average response criterion was positive in both conditions (1.03 ± 0.52 vs 0.36 ± 0.18 for 16-channel vs 1-channel speech, M ± SD across participants). This criterion was also significantly higher for intelligible 16-channel than for unintelligible 1-channel speech ($t(5) = 4.08$, $p = 0.01$, Cohen's d = 1.67). These results demonstrate that participants' ability to detect an irregularity in the stimulus rhythm (reflected in d-prime) was significantly enhanced when speech sequences were presented with greater numbers of vocoder channels and hence were intelligible. This intelligibility-associated enhancement of rhythm perception was driven by a reduced number of false alarms (indicated by a positive response criterion $c$; see Statistical Analysis).

## Experiment 2: Effect of spectral complexity

The findings from our original study [26] and Experiment 1 are consistent with an important role of intelligibility in this rhythmic irregularity detection task, and by extension, in the perception of auditory rhythm. However, 16-channel speech is not only more intelligible than 1-channel speech, but also more spectrally complex. Whereas for 1-channel speech the same broadband amplitude envelope is applied to all frequencies, for 16-channel speech different

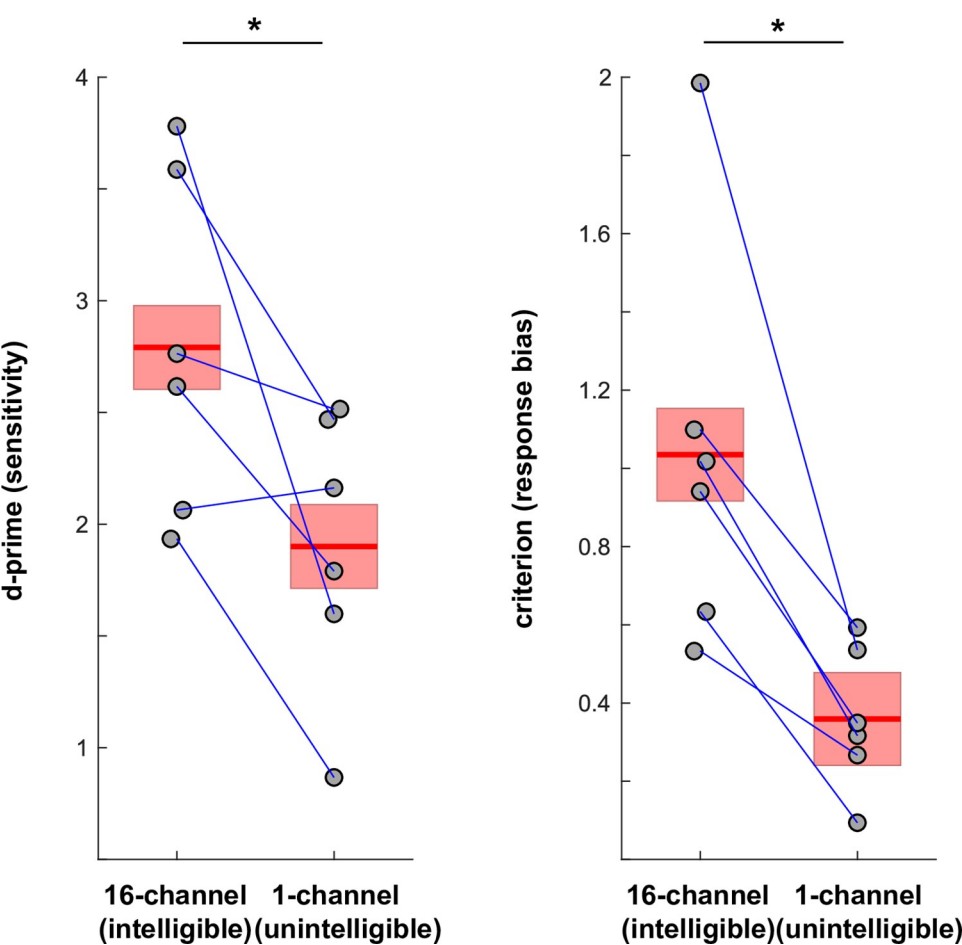

**Fig 2. Sensitivity (d-prime) and response bias (criterion) scores from the Experiment 1 irregularity detection task.**
Points represent data from individual participants/conditions, and lines connect data from the same participant. Mean
and standard error of the mean (SEM), corrected for between-participant variation as appropriate for repeated-
measures comparisons [32], are shown as a red line and coloured area, respectively.

amplitude envelopes are applied to different frequencies. 16-channel speech therefore contains
changes in spectral composition over time, and this increase in spectral complexity could
explain the observed difference between 16- and 1-channel speech in Experiment 1. To test this
hypothesis, we introduced a third condition in Experiment 2: 16-channel noise-vocoded speech
that has been spectrally-rotated (henceforth "16-channel rotated"). This condition was created
by applying amplitude envelopes from low frequency analysis channels to high frequency noise,
and vice-versa, resulting in stimuli that are equally spectrally-complex as intelligible 16-channel
speech, but are unintelligible (due to spectral rotation, cf. refs [25, 33]). Therefore, improved
irregularity detection for the intelligible 16-channel versus unintelligible 16-channel rotated
conditions would be consistent with an effect of intelligibility on the perception of rhythm,
whereas differences between the two unintelligible conditions (1-channel and 16-channel
rotated) would suggest that previous findings were due to an effect of spectral complexity.

## Methods

**Participants.**   Twenty-six participants were recruited from Prolific Academic (www.
prolific.co, cf. ref [34]) and completed the experiment online. They gave informed consent by

clicking on a button. Of these participants, four were excluded for failing a test designed to ensure they were wearing headphones (see Procedure). Two participants completed the experiment but were excluded from subsequent data analyses because they performed at chance levels of accuracy in all conditions (see Statistical Analyses). This left data from 20 participants (13 female; M 34.5 years, SD 9.9 years, range 21–53 years) in the analysis. All participants indicated that they were native speakers of English and were paid £6/h for their time (including those whose data were excluded). We did not screen for age-related hearing difficulties or other conditions that can compromise speech perception abilities, as these are unlikely to affect the relative difference between (e.g., intelligible and unintelligible) conditions. However, we did make sure participants could pass a basic hearing test (with their volume set at a comfortable level), were wearing headphones (in a test described below) and could complete the task (based on performance in practice trials, described below). The study was approved by the Cambridge Psychology Research Ethics Committee (application number PRE.2015.132).

The sample size was estimated based on repeated-measures statistics, an effect size of d = 0.6 and power = 0.8. The effect size was estimated conservatively, considering those obtained in previous work [26] (d = 0.42) and Experiment 1 (d = 1.12 for d-prime).

**Materials.** The same set of rhythmic speech sentences (time-compressed to 3.125 Hz) were presented in the same two experimental conditions (16-channel and 1-channel) as in Experiment 1. However, we added the 16-channel rotated condition to our experimental protocol. This condition was created using the same amplitude envelopes as for 16-channel speech (see Experiment 1, Materials), but the envelopes were swapped between high and low frequencies (i.e. spectrally-rotated) before being used to modulate bands of noise. This yields an unintelligible speech condition that is equated with the intelligible 16-channel speech for spectral complexity.

Irregularities were introduced into the rhythmic sequences as described for Experiment 1 (i.e. 50% irregular trials; second, third, or fourth word shifted forwards or backwards by 68 ms with equal probability; see Experiment 1, Materials).

**Procedure.** The experiment (and subsequent experiments) was conducted over the internet using the jsPsych JavaScript library [35] and JATOS [36] experiment management software. Participants provided informed consent before starting the experiment. Mobile phones and tablets were ineligible, which was ensured with a JavaScript-based device check that ran silently at the start of the study. Participants were instructed to ensure that they were in a quiet, distraction-free environment before beginning the task, and to wear headphones.

The experiment began with a sound calibration task to verify that the audio could be heard clearly, and to allow participants to adjust their volume to a comfortable level. The calibration sound was a sentence spoken by an adult male speaker of British English and equated for subjective loudness with the experimental stimuli. Participants heard the sentence repeated in a loop and were asked to adjust the volume on their computer so that they could hear the words at a clear and comfortable level. They were instructed not to adjust the volume on their computer for the remainder of the experiment.

We next conducted a test designed by Woods et al. [37] to ensure that participants were wearing headphones. Participants were asked to listen to three consecutive 1000-ms pure tones at 200 Hz and to judge which of the tones was the quietest. One of these tones was -6 dB quieter than the other tones, and therefore the correct response. The remaining two tones were equal in intensity. Of these two louder tones, one was presented at opposite phases across the two stereo channels. This 'anti-phase' tone is reduced in its perceived loudness when played via loudspeakers, due to the opportunity for the two opposite-phase left and right signals to interact before reaching the listener. However, this perceptual effect does not occur when the tone is heard via headphones. Responses from participants who do not wear

headphones are therefore biased towards selecting these anti-phase tones as the quietest sound, leading to incorrect responses. There were six trials in total, one for each presentation order of the three sounds, and the trials were presented in random order. If participants responded incorrectly in two or more out of six trials, they were asked to repeat the task, and they were excluded from the remainder of the experiment if they failed to meet this criterion a second time.

Participants were then asked to listen to series of tones, and to press the space bar whenever they heard a tone. The tones started at a relatively high intensity (-10 dB relative to the calibration sound) and followed a simple staircase procedure with four reversals in order to provide an estimate of the threshold of participant's hearing (relative to the level set during the calibration task) for that frequency. This procedure was repeated for four different frequencies: 500, 1000, 2000 and 4000 Hz. This data was collected as part of a separate experiment. On average, participants' hearing threshold measures were 58.64 dB (SD: 10.93 dB) below their self-adjusted comfortable volume level.

Participants then received detailed instructions and practice trials for the main rhythmic irregularity detection task. This task was identical to that used in Experiment 1, with participants indicating after each trial whether the sequence was regular or irregular using one of two keys on their computer keyboard. However, instead of noise-vocoded speech, here the practice stimuli consisted of clear speech (five-syllable sequences, but using different words from those presented in the main part of the experiment) or rhythmic sequences of white noise (five 90-ms bursts of noise, presented at the same 3.125 Hz rate), randomly chosen in each practice trial. In the practice trials that contained an irregularity, the size of the shift was ±68 ms for clear speech, and ±34 ms for white noise, making the detection of irregularities relatively easy. This change allowed us to be more certain that the participants understood the task, which was particularly important for remote web-based testing. Participants received feedback on whether or not they responded correctly after each practice trial. They were only able to continue with the main part of the experiment if they responded correctly in at least 12 out of 16 practice trials. In case of failure, they were asked to repeat practice as often as necessary. Out of our 26 participants, one participant had to repeat the practice once, and another participant repeated the practice task twice before proceeding to the main experiment.

After the practice trials, participants completed 60 trials in each of the three conditions (1-channel, 16-channel, 16-channel rotated; Fig 1B–1D). The condition was selected pseudo-randomly in each trial. Every 60 trials, participants were offered a break and told that they could continue with the task by pressing a key when they were ready.

After the main task, a final part of the experiment was designed to verify the success of our intelligibility manipulation in the different conditions, and test for the possibility of perceptual learning [38]. Participants completed the following tasks: (1) free response stimulus description: They were presented with one example stimulus for each condition (always in the order 16-channel rotated, 1-channel, 16-channel) and, after each one, were asked to describe what they heard in their own words by typing their response into a text field. (2) intelligibility frequency rating: Participants were told that all of the sounds were derived from speech and asked to listen again to one example stimulus per condition. For each of the examples, they were asked to answer the multiple-choice question "How often did you understand the words when you heard a sound like this one?" (options: "Always", "Mostly", "Sometimes", "Rarely", "Never"). They answered this question retrospectively about their perception of each type of stimulus at the beginning, middle, and end of the task. (3) word report: Participants listened to one more example sound per condition (always in the order 16-channel rotated, 1-channel, 16-channel). For each sound, they were told that the first word is always "pick" and the last word is always "up" and asked to write the three words in the middle. They were encouraged to guess if they were not sure.

**Statistical analysis.**    To exclude participants who were unable to perform the rhythm perception task at above chance levels, we estimated the 95% confidence intervals for chance d-prime values using a simulation method. We simulated an experiment which consisted of the same total number of trials (60) and number of irregular trials (50%) as in each of the conditions in our actual experiment. In each simulated trial, a response (irregular or regular) was generated randomly. D-prime was calculated for this simulated dataset, and this procedure was repeated 1,000,000 times, yielding a distribution of d-prime values that would be observed based on random guesses in all trials. We extracted the 95% highest d-prime value from this distribution and defined this value as a threshold for chance performance. For this experiment, this threshold was a d-prime of 0.5, and two participants with a d-prime below or equal to 0.5 were excluded from the analysis.

For the remaining twenty participants, performance in the three conditions was quantified using d-prime and criterion, as described for Experiment 1 (Statistical Analysis). Using paired *t*-tests, performance in 16-channel and 1-channel conditions was then compared to test for the presence of the effect observed in Experiment 1. In addition, performance in 16-channel and 16-channel rotated conditions were compared to test for an effect of spectral complexity.

Ratings in the multiple-choice question (intelligibility frequency rating; see Procedure) were coded from 1 to 5, where 1 corresponds to "Never" and 5 to "Always". We used Spearman's rank order correlations to test for relationships between performance (d-prime or criterion) in the irregularity detection task and these ratings, as well as the number of words correctly reported.

## Results

After the experiment, participants correctly reported, on average, 2.30 ± 1.26 out of 3 words from an example sentence in the 16-channel condition, but only 0 and 0.05 ± 0.22 words in the 1-channel and 16-channel rotated condition, respectively. Chance levels of word report accuracy for unintelligible speech (given knowledge of the three sets of 8 words included in the stimulus set) is 0.125 for each word and 0.375 overall. Hence, this indicates that, as intended, the 16-channel speech was highly intelligible and the other two conditions were entirely unintelligible.

When asked to judge retrospectively how often they understood words during the task (intelligibility frequency rating; see Procedure and Statistical Analysis), participants rated (pooled over ratings from the beginning, middle and end of the experiment) 16-channel speech with a median of 4.17 (4 = "mostly", 5 = "always"; 25% quantile: 3.67, 75%: quantile: 5), 1-channel speech with a median of 1 (= "never"; 25% quantile: 1, 75%: quantile: 1.34), and 16-channel rotated speech with a median of 3 (= "sometimes"; 25% quantile: 2.34, 75%: quantile: 4). These ratings did not change systematically throughout the experiment (the difference scores between the beginning and end of the experiment had a median of 0).

As shown in Fig 3A, we replicated the finding from Experiment 1 that participants showed enhanced sensitivity (i.e. greater d-prime) when detecting rhythm irregularities for intelligible 16-channel speech, as compared to unintelligible 1-channel speech ($t(19) = 4.87$, $p = 0.0001$, Cohen's d = 1.09). In addition, we found enhanced sensitivity for 16-channel speech when compared with 16-channel rotated speech, a comparison which has the same overall difference in intelligibility but is matched for spectral complexity ($t(19) = 3.17$, $p = 0.005$, Cohen's d = 0.71).

Nonetheless, we note that some participants rated 16-channel rotated speech as "sometimes" intelligible, and that a substantial number of participants also gave a similar rating of intelligibility for 16-channel speech. It is therefore worth considering whether differences in d-

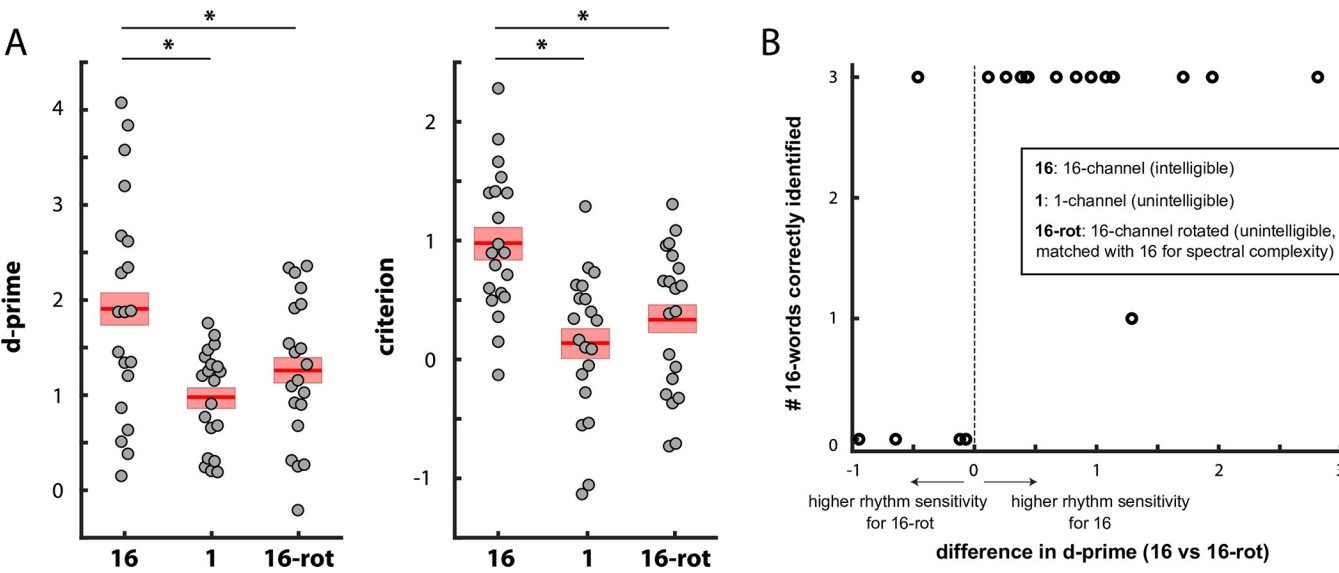

**Fig 3.** A. Sensitivity (d-prime) and response bias (criterion) during irregularity detection in Experiment 2. For other conventions, see caption of Fig 2. B. Number of correctly identified 16-channel vocoded words (between 0 and 3) as a function of the difference in irregularity detection performance (d-prime) between 16-channel and 16-channel rotated speech (shown in A).

prime between 16-channel and 16-channel rotated condition varied as a function of the degree of difference in intelligibility experienced by individual participants. To assess this, we correlated the d-prime difference between 16-channel and 16-channel rotated speech with word report accuracy for these two conditions at the end of the experiment. We observed that d-prime differences were positively correlated with both the number of correctly reported 16-channel vocoded words (word report; see Procedure and Statistical Analysis) after the experiment ($r = 0.52$, $p = 0.018$), and with participants' retrospective intelligibility ratings in the 16-channel condition though this latter effect did not reach significance ($r = 0.43$, $p = 0.06$). Nineteen out of 20 participants reported either none or all of the three 16-channel vocoded words correctly. All four participants who reported no words correctly showed better performance in the 16-channel rotated than in the 16-channel condition (Fig 3B). Fourteen out of fifteen participants who reported all 16-channel words correctly showed enhanced performance in the 16-channel condition. These findings further suggest that greater speech intelligibility–rather than greater spectral complexity–is associated with the ability to detect deviations in speech rhythm.

We also tested whether sensitivity in the irregularity detection task, as well as differences across conditions, changed over the course of the experiment. To do so, we divided trials into quartiles according to their timing during the experiment, and compared performance between those quartiles. We did not find a main effect of quartile on d-prime ($F(3) = 0.44$, $p = 0.73$), nor an interaction between quartile and condition ($F(6) = 0.71$, $p = 0.64$). This indicates that performance was relatively stable throughout the experiment.

As in Experiment 1, the average response criterion was positive in all conditions ($0.98 \pm 0.61$ vs $0.14 \pm 0.62$ vs $0.33 \pm 0.61$ for 16-channel vs 1-channel vs 16-channel rotated speech), and significantly higher for 16-channel speech as compared to both 1-channel speech ($t(19) = 6.09$, $p < 0.0001$, Cohen's d = 1.36) and 16-channel rotated speech ($t(19) = 4.74$, $p = 0.0001$, Cohen's d = 1.06). The difference in criterion between the 16-channel and 16-channel rotated condition was not correlated with any of our measures of intelligibility (all $p > 0.35$).

## Experiment 3: Effect of intelligibility

Experiment 2 shows that, even after controlling for spectral complexity, rhythm perception is enhanced for intelligible compared to unintelligible speech. Nonetheless, our experimental conditions differed in several other acoustic properties which might also explain the observed effect. For example, it has been proposed that the amplitude envelope of certain key frequencies (around 1 kHz) that include vowel formants is the best predictor of the location of rhythmic beats in speech [39]. Energy in these frequencies will be presented at a higher spectral frequency than expected for 16-channel rotated speech. This change might explain impaired rhythm detection in this condition. In our third experiment, we adopted a different approach to the comparison of intelligible and unintelligible speech by contrasting different groups of participants who all listened to the same single set of distorted speech sounds, but whose ability to understand the speech was systematically manipulated by providing or withholding a period of perceptual training.

We used sine-wave speech (Fig 4A) for this purpose, a stimulus created from sets of three sine waves that track the amplitude and frequency of speech formants [40]. Sine-wave speech is of particular interest for this purpose as its intelligibility can be very quickly and dramatically affected by perceptual learning: Naïve listeners, who have never been exposed to sine-wave speech, typically perceive the stimulus as non-speech sounds, such as whistles or bird songs, whereas listeners trained to perceive it as speech can often identify the different words in the stimulus [41]. This property of sine-wave speech has been used previously in, for example, brain imaging experiments to compare brain responses to identical stimuli that are heard as speech or non-speech [43–45].

In this experiment, we thus could compare rhythm perception for acoustically-identical stimuli that were either heard as speech (by trained listeners) or non-speech (by naïve listeners). To achieve this, three different groups of participants performed the rhythmic irregularity detection task on sine-wave speech before and/or after being trained to perceive it as speech. The amount of exposure to sine-wave speech prior to training decreased with each group (Fig 4B). We then contrasted training-induced changes in the ability to detect irregularities in

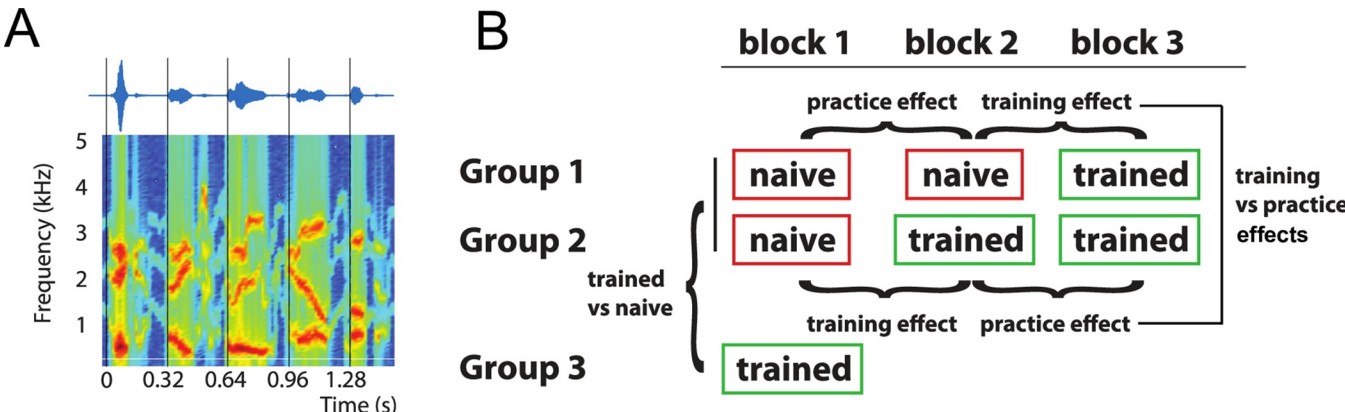

**Fig 4. Stimuli and experimental design in Experiment 3.** A. Sinusoids tracking the first three formants of the original speech (Fig 1A) signal were extracted and combined to form sine-wave speech (see Experiment 3, Materials). B. Three participant groups were trained to perceive the stimulus as speech at different points throughout the experiment (see Experiment 3, Procedure), and we tested how the induced change in perception affected their ability to detect irregularities in the stimulus rhythm. Groups 1 and 2 completed three experimental blocks and were naïve in block 1. Performance in the block immediately following the training, relative to that preceding the training, was defined as the training effect. Performance changes between the two blocks not interrupted by training was defined as the practice effect. Training and practice effects were then compared within and between groups (cf. Fig 6B). Group 3 was trained prior to the experiment and completed only one experimental block. Irregularity detection performance in this group was compared with performance in the other two groups, which were still naïve in block 1 (between-subject design).

rhythmic sine-wave speech with changes due to other factors, such as practice (i.e. improvements in irregularity detection that might occur over time spent doing the task). This was possible by training the three groups at three different time points during the experiment (Fig 4B) so that, at each of these time points, we were able to compare changes in performance in a group that had just undergone training to another which had not. This design requires a larger number of participants than previously since: (1) the critical comparison of naïve and trained performance can only be performed in one order; and (2) each participant can only be tested once (since training effects are long-lasting). By the hypothesis that linguistic properties affect rhythm perception, we would expect improved performance in the irregularity detection tasks when participants perceived the sine-wave stimulus as speech compared to when they were naïve.

Our experimental design included both within- and between participant comparisons. Two participant groups (Groups 1 and 2 in Fig 4B) were trained at later stages of the experiment, i.e. after having completed two or one blocks (i.e. 240 or 120 trials) of the irregularity task as naïve listeners, respectively. For these groups, it was possible to define training effects by contrasting performance in the block following training with that in the block preceding training. Likewise, practice effects were obtained by contrasting performance between successive task blocks not separated by training.

In these two groups, participants were exposed to at least one block of sine-wave speech before they were trained to understand it. Yet, it is possible that some participants spontaneously learnt to perceive the sounds as speech, even before being explicitly taught to do so. This might reduce the significance of within-participant changes caused by the training, leading to smaller or no training effects in one or both groups. We therefore included another participant group (Group 3), which was trained prior to the first experimental block. In this group we were unable to define training effects within participants. However, this enabled us to compare rhythm perception between participants who perceived the stimulus as speech (Group 3) with others who were still as naïve as possible during the first block (Groups 1 and 2), minimizing practice and spontaneous learning effects.

## Methods

**Participants.** 186 participants were recruited via Prolific Academic and completed the experiment online. They gave informed consent by clicking on a button. Participants were assigned to three independent groups of participants (see Procedure). Groups 1 and 2 were tested prior to Group 3, and participants randomly assigned to one of the two groups. Group 3 was tested later, to address potential effects of practice and spontaneous learning in the other two groups (see Experiment 3, Procedure and Statistical Analysis). Eighteen participants (6, 7 and 5 in the three groups) failed the test designed to ensure they were wearing headphones (see Experiment 2, Procedure) and were excluded from the remainder of the experiment. Nine participants (3, 2 and 4 in the three groups) completed the experiment but were excluded from further analyses due to chance performance in all experimental blocks (see Statistical Analysis). Seven participants (3, 4 and 0 in the three groups) were excluded from further analyses due to a monotonic decrease in d-prime of more than 0.4 over all blocks. As the majority of participants showed a strong increase in d-prime, i.e. learning effects, over the course of the experiment (reported in Results), we took this decrease as an indicator of excessive fatigue. However, a re-analysis of the dataset with all participants included did not change the pattern of results reported here.

Fifty-nine participants (36 female; M 35.8 years, SD 10.1 years, range 20–57 years) remained in Group 1 for subsequent data analyses, 60 participants (33 female; M 35.3 years,

SD 9.9 years, range 19–55 years) remained in Group 2, and 33 participants (21 female; M 32.6 years, SD 10.3 years, range 18–61 years) remained in Group 3. All participants indicated that they were native speakers of English and were paid £6/h for their time (including those who were excluded). The study was approved by the Cambridge Psychology Research Ethics Committee (application number PRE.2015.132).

The sample size for Groups 1 and 2 (who were tested first) was estimated based on between-subject statistics, an effect size of d = 0.45 and power = 0.8. The effect size was estimated to be lower than that observed in Experiments 1 and 2 (1.12 and 1.09, respectively, for d-prime). This is because differences in acoustic properties between intelligible and unintelligible speech, that were present in those experiments and might have contributed to differences in speech rhythm perception, were eliminated in Experiment 3 (which compares groups of participants exposed to identical acoustic stimulation but with hypothesized differences in speech perception). As these two groups (totalling 119 participants) provided us with a very reliable estimate of speech rhythm perception in naïve participants (in their first experimental block), we halved the sample size for Group 3 (for the comparison of performance between naïve and trained participants in the first block; see Statistical Analysis).

**Materials.** The same rhythmic sentences of clear speech (spoken at 2 Hz; Fig 1A) were used as described for Experiment 1 (Materials). For each original sentence, sinusoids were created that track the first three formants of the speech signal [40] and combined to form sine-wave speech (while discarding the rest of the speech signal). This was done using Praat software (version 6.12, from http://www.fon.hum.uva.nl/praat/download_win.html) and a script written by Chris Darwin (http://www.lifesci.sussex.ac.uk/home/Chris_Darwin/Praatscripts/SWS). While it is well established that the intelligibility of algorithmically-constructed sine-wave speech is inferior to that constructed by hand [41], for our purposes this script was sufficient to create a stimulus that is intelligible to informed listeners, but not to naïve listeners. The resulting sine-wave speech sentences (Fig 4A) were time-compressed to 3.125 Hz using the PSOLA algorithm implemented in Praat.

Irregularities were introduced into the rhythmic sequences as described for Experiment 1 (i.e. 50% irregular trials, in which the second, third, or fourth word was shifted forwards or backwards with equal probability; see Experiment 1, Materials).

**Procedure.** Participants were informed that the stimuli in the experiment would consist of computer-generated sounds without mentioning speech. The initial steps (consent, sound check, headphones test, practice) were carried out as described for the procedure of Experiment 2.

Similar to Experiment 2, participants' hearing threshold was estimated as 66.19 dB +/-11.84 dB (mean +/-SD) below their self-adjusted comfortable volume level.

To avoid exposure to similar (i.e. rhythmic, one-syllable) speech sounds before the main part of the experiment, which might influence the effect of training, we used 500-Hz pure tones for the practice stimuli (for which we used a smaller shift size of ±34 ms). Participants continued with the main task if they responded correctly in at least 6 out of 8 practice trials (across all groups, two participants repeated the practice trials once, and two participants twice).

After the practice trials, participants completed the main irregularity detection task which was divided into blocks of 120 trials (with an additional break after 60 trials in each block). Group 1 and 2 completed three blocks and Group 3 only completed one block. The irregularity detection task was interrupted by a short training session, designed to give participants perceptual insight such that they could then understand the sine-wave speech. The three experimental groups were trained at different times within the experiment (shown in Fig 1C), with the amount of exposure to sine-wave speech prior to training decreasing with each group:

Training occurred preceding block 3 (Group 1), preceding block 2 (Group 2), or preceding block 1 (i.e. immediately after practice and before the main task; Group 3). During training, participants were first informed that the presented sounds are derived from human speech. They then listened to five example stimuli which each consisted of clear speech followed by the sine-wave version of the same sentence. In previous studies [43–46], similar approaches have been successful in training listeners to perceive sine-wave speech as speech (see ref [42], for discussion).

In addition to the main task and training, participants completed another short test (stimulus description task) at various time points throughout the experiment, designed to covertly assess their ability to understand sine-wave speech without revealing that it is speech. This test started with the presentation of one sine-wave speech sequence (consisting of five one-syllable words), followed by the multiple-choice question "Choose the item in the list below that you think describes the sound best" (responses options were offered in randomized order: "Bird Sounds", "Several Simultaneous Sounds", "SciFi Sounds", "Alien Language", "Music", "Radio Interference", "Computer Beeps", "Whistles", "Speech", or "Other"; see ref [40]). If participants selected "other", a textbox popped up in which they were asked to elaborate. If participants selected the "speech" option, they were asked to listen to another example sine-wave speech sequence and answer the multiple-choice question "How many words could you have repeated aloud from the last sound that you heard" (options: "All", "Most", "Some", "A few", "None"). Participants completed this test before and after training, and in addition, after the first block (Groups 1 and 3) or after the second block (Group 2), as illustrated in Fig 5. In this way, we were able to measure how the perception of sine-wave speech develops over the course of the experiment, and how training affects this perception.

After the main task, participants completed the following tasks in a final debrief, similar to that described for Experiment 2. (1) Intelligibility frequency rating: They were asked to answer the multiple-choice question "At the following time points during the task, how often could you have repeated aloud words from the speech that you heard?" (options: "Always", "Mostly", "Sometimes", "Rarely", "Never"), for the time points "beginning of task", "before you were trained to identify the sounds as speech", "after you were trained to identify the sounds as speech", and "end of task". (2) Word report: Participants listened to three sine-wave speech sequences, and after each sentence, were asked to type the three middle words into a text box (they were told that the first word is always "pick" and the last word is always "up").

**Statistical analysis.**   Performance in the irregularity detection task was quantified using d-prime and criterion $c$, as described for Experiment 1 (Statistical Analysis). This was done separately for each experimental block and the three groups of participants. Chance level was simulated as described for Experiment 2 (Statistical Analysis), based on the number of trials in each block (120). These simulations yielded a d-prime of 0.34 as the 95% confidence interval threshold for chance performance. Participants with a d-prime below or equal to this level in each block were excluded from the study (3, 2 and 4 in the three groups). The following statistical analysis then involved both within- and between-participant comparisons (Fig 4B).

For Groups 1 and 2, training effects were calculated by subtracting individual d-prime and criterion values during the test block preceding training (block 2 in Group 1, block 1 in Group 2) from those resulting from the block following training (block 3 in Group 1, block 2 in Group 2). Similarly, we calculated practice effects (i.e. changes in performance that are due to participants practicing the irregularity detection task and therefore unrelated to intelligibility changes) as the performance differences between blocks that were not separated by training (i.e. block 2–1 in Group 1, block 3–2 in Group 2). These performance differences were then compared in a mixed ANOVA with one within-subjects factor (training vs practice) and one between-subjects factor (earlier training, i.e. Group 2, vs later training, i.e. Group 1).

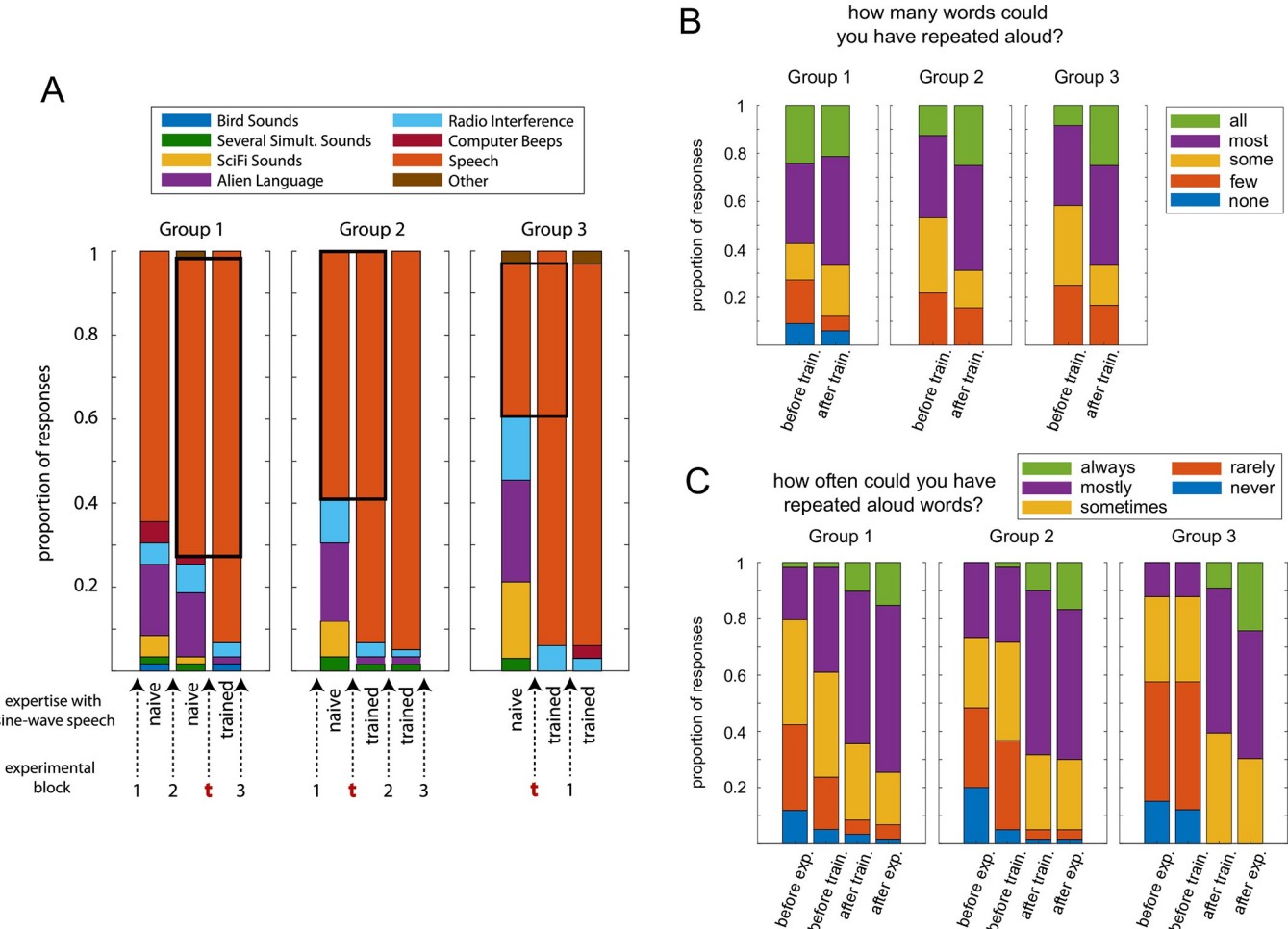

**Fig 5. Testing the perception of sine-wave speech throughout the experiment.** A. Proportion of responses given in the three different groups when asked to describe the sine-wave speech (multiple-choice stimulus description task; see Experiment 3, Procedure). Note that all groups completed this questionnaire before and after training and, in addition, after the second (Group 2) or first experimental block (Groups 1 and 3). All groups therefore completed three questionnaires, even though Group 3 only completed one experimental block. The bottom part shows the experimental course with training (t), the experimental blocks, and assumed expertise with sine-wave speech. B. Proportion of responses to the question "How many words could you have repeated aloud?", presented immediately after participants selected "speech" in the stimulus description task. Only data from participants is shown whose response was "speech" both before and after training in A (black squares). C. Proportion of responses to the question "How often could you have repeated aloud words from the speech you heard?", answered by all participants after the experiment, retrospectively about various points during the experiment (intelligibility frequency rating; see Experiment 3, Procedure).

As Group 3 was trained prior to the first experimental block, we were unable to contrast performance before and after training within participants, as in the other two groups. However, we compared performance in the first block with the average performance of the other two groups in the same block (using a between-group *t*-test). This approach minimized practice and implicit learning effects in the naïve groups (Groups 1 and 2; Fig 4B).

If participants described sine-wave speech as "other" in the stimulus description task (see Procedure), their explanatory text response was evaluated post-hoc and counted as the "speech" option if it was sufficiently close (e.g., "computer-distorted speech"). This was the case in fourteen out of the seventeen "other" responses throughout the experiments and across groups. Responses to the multiple-choice question (intelligibility frequency rating; Procedure) were coded from 1 to 5, where 1 corresponds to "None" or "Never" and 5 to "All" or "Always", respectively.

We examined the relationship between performance in the irregularity detection task (d-prime or criterion) and the number of correct items in the word report task (between 0 and 9; Procedure) using Pearson's correlation.

## Results

Fig 5A shows how participants in the three groups described the sine-wave speech at various time points during the experiment (stimulus description task; see Procedure). Although a relatively large proportion of participants spontaneously described the stimulus as speech, this proportion increased after training by 22.0%, 32.2%, and 57.5% in Groups 1, 2, and 3, respectively (based on responses immediately after vs immediately before training, see Procedure). This increase was significant in all three groups (Group 1: $\chi^2$ = 13.08; Group 2: $\chi^2$ = 17.05; Group 3: $\chi^2$ = 17.05; $p < 0.0001$ in all groups; McNemar's test) and is in line with earlier studies [43–46]. Almost all participants perceived the stimulus as speech immediately after training (93.2% vs 93.2% vs 93.9% in Groups 1–3), with no significant difference among the groups ($\chi^2$ = 0.02, $p = 0.99$; chi-squared test). However, there was a significant difference among groups immediately before training ($\chi^2$ = 10.74, $p = 0.005$), with more participants describing the stimulus as speech in the groups that were trained later (71.2% vs 61.0% vs 36.4% in Groups 1–3).

In the groups trained relatively early, the participants who described the stimulus as speech before and after training rated it as more (although not significantly more) intelligible after than before training (Group 3: median of 3 vs 4 for before vs after training, corresponding to "some" and "most", respectively; $z = 1.20$, $p = 0.23$, Wilcoxon rank sum test; Group 2: median of 3 vs 4; $z = 1.68$, $p = 0.09$; the complete distribution of responses is shown in Fig 5B). This was not the case in Group 1, which was trained latest (median of 4 vs 4; $p = 0.54$). When asked to rate stimulus intelligibility retrospectively (intelligibility frequency rating; Fig 5C), the groups differed significantly when referring to immediately before training ($\chi^2$ = 12.02, $p = 0.003$; Kruskal-Wallis test by ranks), with lowest intelligibility rating in the group trained first (Group 3 vs Group 2: $z = 2.26$, $p = 0.02$; Group 3 vs Group 1: $z = 3.42$, $p = 0.0006$; Wilcoxon rank sum test). Ratings did not differ significantly between groups at other time points ($p > 0.42$). Together, results indicate that, in addition to a pronounced training effect on the intelligibility of sine-wave speech, participants are more likely to spontaneously identify the stimulus as speech when they are exposed to it for an extended period of time without explicit training.

Fig 6 shows how training and practice affected the detection of irregularities in stimulus rhythm. We first defined training and practice effects for Groups 1 and 2, who completed three experimental blocks (i.e. performance changes between two blocks with or without training in between, respectively; see Statistical Analysis and Figs 4B and 6A). For d-prime, a mixed (within- and between-subjects) ANOVA yielded a significant interaction ($F(1) = 5.10$, $p = 0.03$) between the within-subjects factor (training effect vs practice effect) and between-subjects factor (earlier training in Group 2 vs later training in Group 1). Post-hoc $t$-tests (Fig 6B) indicated that training to understand sine-wave speech improved rhythm perception significantly more than practice, but only in the group that was trained earlier during the experiment (Group 2: $t(59) = 2.23$, $p = 0.03$, Cohen's d = 0.29; Group 1: $t(58) = 0.90$, $p = 0.37$, Cohen's d = 0.12; Fig 6B).

In principle, this result can be explained by practice effects that are strongest in the early parts of the experiment and therefore produce a change in performance that resembles the training effect, but only in the group that is trained relatively early (Group 2). Alternatively, the relatively large improvement in performance prior to training in Group 1 can also be

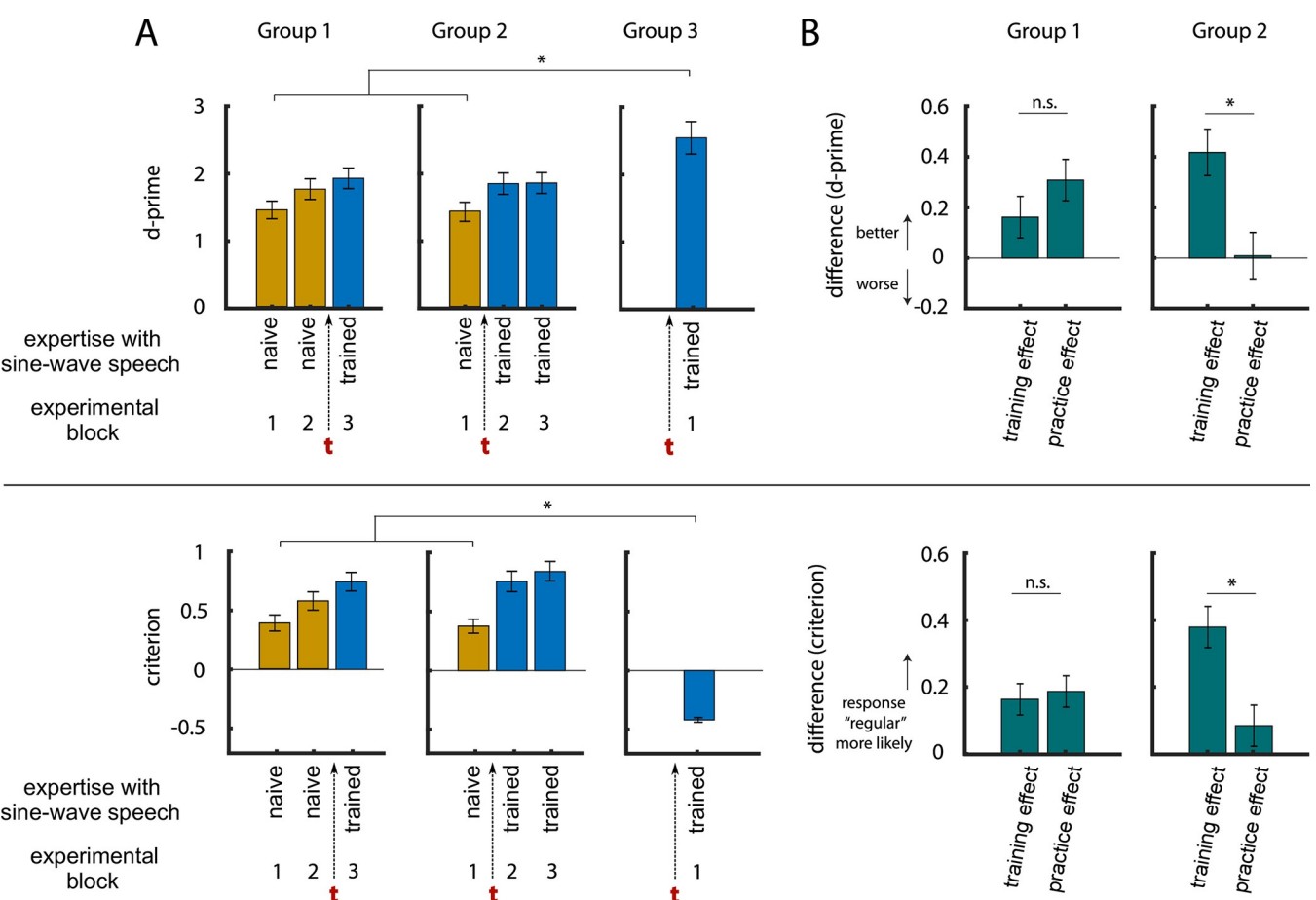

**Fig 6.** A. Sensitivity (d-prime, top) and response bias (criterion, bottom) in the three groups during irregularity detection in Experiment 3, and how they develop with expertise with sine-wave speech. The red t shows when participants completed training. Error bars show SEM, corrected for between-participant variation. B. In Groups 1 and 2, some of the experimental blocks (shown in A) were contrasted to reveal training (block 3–2 in Group 1, block 2–1 in Group 2) and practice effects (block 2–1 in Group 1, block 3–2 in Group 2), respectively.

explained by participants spontaneously learning to identify the stimulus as speech (and therefore be related to intelligibility), as hypothesized above. In order to obtain a better estimate of the training effect before practice-related improvements and/or spontaneous learning had the chance to occur, we also compared performance in the first experimental block only between naïve participants (Groups 1 and 2) and those who underwent training prior to that block (Group 3). We found that trained participants performed significantly better than naïve participants ($t(150) = 4.95$, $p < 0.0001$, Cohen's d = 0.65). This improvement in speech rhythm perception cannot be explained by practice effects, as all groups spent an equal amount of time doing the task. We can therefore conclude that the improved speech rhythm perception during the first block of trials in Group 3 is a direct consequence of sine-wave speech being more intelligible to these participants.

For the response criterion, a mixed ANOVA also yielded a significant interaction ($F(1) = 4.21$, $p = 0.04$), with post-hoc $t$-tests indicating a significantly increased criterion induced by training only in Group 2 (Group 2: $t(59) = 2.39$, $p = 0.02$, Cohen's d = 0.31; Group 1: $t(58) = 0.25$, $p = 0.80$, Cohen's d = 0.03). In the first experimental block, trained participants (Group 3) showed a significantly smaller response criterion than naïve participants in the other two groups ($t(150) = 9.35$, $p < 0.0001$, Cohen's d = 1.15).

Neither training- nor practice-induced changes in d-prime or criterion (Fig 6B) were correlated with the number of correctly reported words after the experiment (all $p > 0.30$). This result is perhaps unsurprising, given the large changes in d-prime and criterion throughout the experiment, and the delay between when training and practice effects are measured and when the word report task is administered.

## General discussion

### Summary

In a series of experiments, we demonstrated that human perception of acoustic rhythms is enhanced by the ability to perceive and identify these sounds as speech (i.e. intelligibility). In a previous study, we had reported that the detection of a temporal irregularity in rhythmic vocoded speech is enhanced if it is vocoded with sufficient channels to be intelligible [26] (16- vs 1-channel vocoded). In Experiment 1, we replicated this effect (1) with a greater proportion of irregular trials (50% instead of 14%), (2) during a forced-choice (irregular/regular) paradigm, and (3) in the more favourable listening conditions to be found outside of an MRI scanner (Fig 2).

In Experiment 2, we replicated this effect again, this time using a web-based implementation of the experiment (Fig 3A). We included an additional 16-channel rotated speech condition, which is unintelligible but has the same spectral complexity as intelligible (non-rotated) 16-channel speech. Since this unintelligible condition also showed reduced irregularity detection performance compared to intelligible 16-channel speech, we can conclude that spectral complexity is not the cause of the difference in rhythm perception. We further found a correlation between individual differences in irregularity detection (intelligible 16-channel vs unintelligible 16-channel rotated) and participants' ability to report words from the intelligible condition (Fig 3B). In principle, increased effort or motivation in some participants could lead to both increased performance in both word report and irregularity detection, and thus to an indirect correlation between the two. The fact that we used the within-subject difference between conditions for the correlation rules out this possibility, as the relative difference between conditions is unlikely to be affected by such general factors. These results are consistent with the proposal that speech intelligibility modulates the perception of speech rhythm. However, there remain other acoustic differences between 16-channel and 16-channel rotated speech that might still explain these findings, e.g. different frequency bands will show greater amplitude modulation or co-modulation. This reflects a more general problem for studying the potential effect of intelligibility; there are a great many acoustic confounds that could explain differences between speech and non-speech conditions.

In Experiment 3, we eliminated any acoustic differences between conditions and instead manipulated participants' perception of a single set of sine-wave speech stimuli, by training groups of participants to perceive the stimulus as speech at different points throughout the experiment. Given the relatively large proportion of participants that spontaneously identified this stimulus as speech, it is likely that the training enhanced speech perception rather than changing it categorically. Nonetheless, the effect of training was to substantially modify speech intelligibility without changing the acoustic form of our sine-wave speech stimuli–which achieves our empirical goals. We demonstrated that participants who perceive sine-wave speech as speech-like (i.e. more intelligible) also show an increased sensitivity to deviations from the stimulus rhythm (Fig 6). This effect cannot be explained by acoustic differences or other stimulus properties, since identical stimuli were used in the intelligible (trained) and unintelligible (naïve) conditions. Hence, we can conclude that perception of auditory rhythm is guided by the perceived timing of linguistic units which can be more accurately perceived

for intelligible speech. Interestingly, in all of these experiments, with the exception of Group 3 in Experiment 3, the observed improvements in sensitivity to detect rhythmic irregularities (quantified using d-prime) were due to a reduced number of false alarms for intelligible speech, reflected by a positive response criterion.

## Linguistic properties contribute to auditory rhythm perception

We emphasize that our study was not designed to answer the long-standing debate on the importance of rhythm for the perception of speech [4–6, 12–14]. Rather, we tested the role of intelligibility for the perception of rhythm in speech that is already rhythmic (in terms of both production and perception, i.e. produced with the p-centre of every syllable aligned with an inaudible metronome beat). In other words, we show that making rhythmically-spoken speech intelligible improves the perception of this rhythm, but not that the rhythm of speech is important for its intelligibility. This result does, of course, *not* imply that acoustic properties–such as amplitude modulations or spectral detail–do not contribute to speech rhythm perception. Rather, it implies that these acoustic cues are not the only factors determining rhythm perception. Our results suggest an important role of linguistic properties which are associated with, but not fully reducible to, acoustic properties; these linguistic properties can only be accessed by listeners when speech is intelligible and lead to the improved perception of speech rhythm.

Our results also provide some evidence of which acoustic properties might be most closely associated with rhythm perception. For example, we consistently found improved rhythm perception for 16-channel as compared to 1-channel speech. This finding is interesting because in 1-channel, but not 16-channel speech, all spectral frequencies are modulated by the same amplitude envelope (i.e. they are perfectly co-modulated, as shown by the stable patterns over frequency in Fig 1B). Hence 1-channel speech has more consistent amplitude fluctuations, yet our results show that it does not lead to a more stable perception of stimulus rhythm. Therefore, stimulus properties other than broadband amplitude fluctuations likely play a more important role for the perception of speech rhythm. Based on results in the 16-channel rotated condition, a mere increase in spectral detail does not seem critical either; instead, speech rhythm perception might be supported by certain spectro-temporal patterns, i.e. acoustic information that is present at a specific frequency and time (cf. ref [47]). Interestingly, several studies have suggested that amplitude fluctuations in a frequency band around 1 kHz are most important for intelligibility, despite not containing most energy in the speech signal [48, 49]. It is possible that this frequency band is also critical for a reliable extraction of speech rhythm. This might explain why participants showed impaired rhythm perception during 1-channel speech and 16-channel rotated speech: If amplitude fluctuations in this frequency band are distorted during the construction of these stimuli, this might lead to similarly distorted p-centres of individual words (i.e. they might not be equally spaced in time anymore), and thus to an erroneous perception of rhythm.

Such acoustic properties however–even those specific to intelligible speech–cannot explain results observed in Experiment 3. We see changes in rhythm perception for stimuli that exclusively differ in their linguistic percept but not in any of their acoustic properties. This finding is in line with previous work concluding that the acoustic correlates of the p-centre are not sufficient to explain their operation [9, 11]. Indeed, it has been suggested that p-centres arise from knowledge about how a speech sequence is created [50, 51]. According to this view, the perceptual events that determine the rhythm of a stimulus are defined, not based on their acoustics, but rather based on the action producing the sound, or on underlying, abstract properties of that stimulus. Our results are in fundamental agreement with this notion, adding further evidence that linguistic properties are critical for how listeners extract p-centres from

human speech: It is possible that intelligibility makes it easier for the listener to separate the "signal from the noise", that is, the ability to translate complex acoustic information into a series of words helps the listener reduce the perceived sounds to p-centers and ignore irrelevant acoustic information. If linguistic properties are absent and information cannot be mapped onto phonemes or other abstract units, this might lead to an unreliable internal representation or storage of stimulus rhythm and, consequently, to poorer performance in the irregularity detection task. In the current study, all native speakers of English were eligible to participate, irrespective of their proficiency in other languages (e.g., bilingualism) or other factors. It would be interesting to test whether differences in rhythmic patterns between languages (e.g., differences between English and Spanish rhythm as carried by duration cues [52]) can influence results reported here. Other future work could examine whether and how the location of p-centres changes when speech becomes unintelligible, and how our results relate to the perception of rhythm in natural speech.

## How do linguistic properties improve auditory rhythm perception?

We here offer mechanistic explanations of how access to these linguistic properties might improve perception of rhythm. These explanations remain speculative and need to be explored in future studies. First, much evidence has highlighted stages of the human auditory system that appear to be specialized for processing human speech; for example, this is shown by sensitivity to amplitude modulation peaks in a frequency range that is typical for human speech [15]. Brain imaging studies revealed that "higher-level" auditory regions, such as Superior Temporal Gyrus (STG) or Sulcus (STS) respond more strongly to intelligible than to unintelligible noise-vocoded speech [24] and seem "tuned" to spectro-temporal patterns that are specific to speech [47]. Many of the same regions in posterior STG seem to be specifically engaged when sine-wave speech is made intelligible by prior knowledge [43, 45].

A second independent line of research that supports similar conclusions comes from the field of "neural entrainment" [53, 54]. This phenomenon describes rhythmic neural activity, typically termed "neural oscillations", that are aligned with rhythmic stimuli, including speech [55–57]. Interestingly, neural entrainment is also stronger for intelligible than unintelligible speech [3, 23, 58, 59], and intelligible speech seems to more reliably entrain endogenous neural oscillations [60]. The manipulation of entrainment using transcranial brain stimulation can change speech processing [61–63], but both immediate and sustained effects seem specific to intelligible speech [26, 60].

Thus, both brain imaging studies and studies of neural oscillations converge on the proposal of speech-specific processing in the human auditory system. Such speech-specificity might underlie the improved rhythm perception for intelligible speech in the current study. Access to linguistic properties–driven by acoustic features that support speech identification– might be necessary to activate speech-specific brain regions and corresponding resources for stimulus processing. These regions might be important for the prediction of upcoming linguistic information [28, 64], including its timing, and therefore sensitive to violations of these predictions. Neural oscillations have been linked to these predictive processes [65, 66], and represent a potential neural substrate of our observed effects.

Although the implication of the motor system in speech perception remains much debated [67–69], there is increasing evidence for an important role of motor regions for time perception [70–72]; a process that again may involve neural oscillations [73–75]. Interestingly, a recent study [76] measured an oscillatory "footprint" of the motor system (mu/beta suppression, reflecting engagement of the motor system) and reported effects (stronger mu/beta suppression during accurate speech perception) that are specific to speech and not present for

auditory control stimuli. These findings link to a range of existing functional imaging evidence for engagement of motor regions during active speech perception [77, 78] and representation of abstract linguistic information (e.g. articulatory gestures or phonemes) in these brain regions [79–81]. If the motor system–or at least its part that is involved in temporal predictions–is indeed more active during perception of intelligible speech, then it represents a promising candidate for future investigations of anatomical and functional substrates of speech rhythm perception.

Our study suggests a potential link between speech rhythm and the literature on "beat-based timing" in rhythm perception. Previous rhythm perception studies have, thus far, typically focused on simple stimuli such as trains of clicks or tones [72]. A network comprising basal ganglia, thalamus and cortical (mostly motor-related) areas seems to be critical for the human ability to perform beat-based timing tasks, during which deviations from isochrony are detected [82–84]. It is not fully clear whether similar regions are also important for the extraction of rhythm in speech sounds, although some studies support this notion [68, 85]. One interpretation of motor system involvement in speech perception [77, 78] is that access to motor representations critically distinguishes perception of speech from non-speech. In the present study, our definition of "intelligibility" is similarly linked to the ability to access motor representations from speech signals (i.e. intelligibility is measured or rated based on how many words listeners can repeat). Hence, enhanced detection of rhythmic deviations for intelligible speech could reflect a beneficial effect of access to motor, beat-based mechanisms for intelligible speech. However, further research will be required to determine whether this, or an alternative "duration-based timing" mechanism is responsible for the present findings. This latter mechanism allows the absolute duration of an interval to be judged without an underlying beat, and seems to rely on a different underlying brain network centred on the cerebellum [82–84].

## Conclusion

The sequence of experiments reported here demonstrate that there are differences in rhythm perception which can only be explained by linguistic (and not acoustic) properties of speech. In Experiment 1 and 2, we ruled out salient speech vs non-speech acoustic differences as responsible for the observed advantage of intelligible speech for accurate rhythm perception (broadband envelope changes in Experiment 1, spectral complexity in Experiment 2). Using acoustically-identical stimuli that are perceived as speech after training, in Experiment 3 we ruled out all remaining acoustic factors as explaining differences in rhythm perception between speech and non-speech.

Our study demonstrates that linguistic properties that are uniquely accessible in intelligible speech contribute to auditory rhythm perception. This finding supports the notion that rhythm perception is influenced by the listener's broader knowledge and experience with the abstract sound patterns that are being generated [50, 51] or, perhaps more precisely, the sounds that the listener knows the speaker intended to generate. Further studies are needed to confirm or falsify different hypotheses concerning the functional and neural underpinnings of this effect.

## Acknowledgments

The authors thank Kevin Woods for providing the stimuli for the headphone test. They are also grateful to Elisa Filevich, Kristian Lange, and Josh De Leeuw, for their work on the development or JATOS and jsPsych libraries, respectively.

## Author Contributions

**Conceptualization:** Benedikt Zoefel, Rebecca A. Gilbert, Matthew H. Davis.

**Data curation:** Benedikt Zoefel, Rebecca A. Gilbert.

**Formal analysis:** Benedikt Zoefel.

**Funding acquisition:** Benedikt Zoefel, Matthew H. Davis.

**Investigation:** Benedikt Zoefel.

**Methodology:** Benedikt Zoefel, Rebecca A. Gilbert.

**Project administration:** Benedikt Zoefel, Matthew H. Davis.

**Resources:** Rebecca A. Gilbert.

**Software:** Rebecca A. Gilbert.

**Supervision:** Matthew H. Davis.

**Validation:** Matthew H. Davis.

**Visualization:** Benedikt Zoefel.

**Writing – original draft:** Benedikt Zoefel.

**Writing – review & editing:** Benedikt Zoefel, Rebecca A. Gilbert, Matthew H. Davis.

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
