## [Decision Letter · Decision Letter 0]

16 Aug 2022

PONE-D-22-14693Intelligibility improves the perception of speech rhythmPLOS ONE

Dear Dr. Zoefel,

Thank you for submitting your manuscript to PLOS ONE. After careful consideration, we feel that it has merit but does not fully meet PLOS ONE’s publication criteria as it currently stands. Therefore, we invite you to submit a revised version of the manuscript that addresses the points raised during the review process. Both reviewers had positive things to say about the manuscript, but I agree that greater clarity in definitions and some further explication around the rationale and interpretations would be useful. In addition, further support for some experimental interpretations (either experimental support or further support from existing literature) would be helpful.

We look forward to receiving your revised manuscript.

Kind regards,

Jessica Adrienne Grahn

Academic Editor

PLOS ONE

Journal Requirements:

6. Please expand the acronym “CNRS” (as indicated in your financial disclosure) so that it states the name of your funders in full.

Reviewers' comments:

Reviewer's Responses to Questions

**Comments to the Author**

1. Is the manuscript technically sound, and do the data support the conclusions?

Reviewer #1: Yes

Reviewer #2: No

2. Has the statistical analysis been performed appropriately and rigorously? 

Reviewer #1: Yes

Reviewer #2: No

3. Have the authors made all data underlying the findings in their manuscript fully available?

Reviewer #1: Yes

Reviewer #2: No

4. Is the manuscript presented in an intelligible fashion and written in standard English?

Reviewer #1: Yes

Reviewer #2: Yes

5. Review Comments to the Author

Reviewer #1: ### GENERAL REMARKS ###

In this manuscript, the authors investigate whether or not intelligibility impacts listeners’ ability to detect rhythmic irregularities in speech. They report the results of three experiments. In the first experiment, they replicate their previous finding that irregularity detection is better for intelligible (16-channel noise-vocoded) compared to unintelligible (1-channel noise-vocoded) speech. In the second experiment, they rule out the possibility that this result was due to enhanced spectral complexity by demonstrating better irregularity detection for intelligible 16-channel noise-vocoded speech compared to largely unintelligible 16-channel spectrally-rotated noise-vocoded speech. In the third experiment, they attempt to remove any possible acoustic confounds by using sine-wave speech stimuli and comparing irregularity detection when listeners were trained or untrained in perceiving the stimuli as speech. Results again support the idea that intelligibility aids irregularity detection.

I very much enjoyed reading the paper: it is clear, well-written and compelling. The results are thoughtfully discussed in the light of existing findings/theories and plausible hypotheses are generated about potential underlying mechanisms. The authors also highlight some possible implications for motor-based theories of speech perception. Additionally, and as a proponent of online testing, I was heartened to see replication of results obtained in the lab via an internet-based implementation.

My comments are mostly very minor requests for clarification or additional information.

### MAJOR COMMENTS ###

1. Carry-over effects. If I’ve understood correctly, then the stimuli were always presented at the same rate. If that’s the case, I wonder whether there were “carry-over effects” from trial to trial. In other words, did listeners gradually build up increasingly strong expectations about speech rate, thus increasing their sensitivity to irregularities over the course of each experiment? Or was there some kind of perceptual “reset” between items, meaning that listeners started each trial with a blank slate, temporally-speaking? N.B. This is different to the “practice effect” identified in Exp 3: in that case, the issue was whether listeners had started to hear the stimuli as speech through mere exposure; in this case, the build-up of temporal expectations could function independently of intelligibility. Having said that, though, I could easily imagine that intelligibility might differentially affect performance for early vs. late trials if there is indeed a build-up of temporal expectations over time. I appreciate the need to pool data for the sake of the SDT-based statistical analyses, but it would be interesting to see if sensitivity changed over time for the intelligible vs. unintelligible conditions of Exps 1 and 2 (by comparing e.g. early vs. late trials).

2. Experiment 3 stimuli. Thanks to the authors for making example stimuli available via the OSF! Having listened to these, I have to say that the sine-wave speech stimuli used in Exp 3 already sound very speech-like (and indeed intelligible) to me (although I do have experience working with SWS). More to the point, most participants already seem to be identifying the SWS as speech from very early on (Fig. 5A). I appreciate that the effect of interest in Exp 3 is actually the training/practice effect, which is clearly contributing to improved irregularity detection. However, I wonder if the authors could be clearer that this improved sensitivity is due to an *enhancement* of (a likely already existing) perception of speech/intelligibility rather than a binary distinction between perceiving the stimuli as “speech” or “not speech” (this binary distinction being strongly implied by some phrasing e.g. p38, line 784). In other words, there’s a continuum when it comes to the perception of speech-like-ness/intelligibility in the stimuli, and a shift along this continuum seems to affect the perception of rhythmic regularity.

3. Experiment 3 sample. I understand the authors’ motivation for recruiting a smaller sample size for Group 3 (lines 568-569), but is there a reason for the specific sample size chosen? It would be good if the authors could be explicit about this (even if it was dictated by time/finances etc.!).

4. Experiment 3 training. It seems a little misleading to describe the training as “successful” on the basis that more participants described the stimuli as speech, given that the training explicitly tells participants that the stimuli were derived from speech! (p34, lines 683-4.) Would it not be better to examine the relationship between whether or not participants described the stimuli as speech and how they performed on the word reporting task? I also wonder about individual differences: my experience with SWS training is that it’s rather binary – some participants learn it almost immediately and others never quite get it. Could the authors comment on individual differences in their data in terms of whether and to what extent training and/or practice led to changes in irregularity detection?

5. Experiments 2 and 3 online recruitment and procedures. a) The authors state that participants were recruited via Prolific and were “native speakers of English”. I’m curious as to why other Prolific filters were not used. In particular, I wonder whether the authors considered restricting the sample to monolingual English speakers (given potential differences in speech rhythm perception across languages) and/or restricting the age range (given potential age-related changes in speech perception abilities). Prolific also has a filter for self-reported hearing loss or hearing difficulties, which may have been worth applying here. b) I appreciate that the data from the pseudo-audiogram procedure (described on p21, lines 355-361) were collected for a different experiment, but might it be possible to report the data here nevertheless, to give the reader some indication of the participants’ hearing sensitivity (especially given the extended age range, as already mentioned)? c) What was the “device check” (p20, line 331) that detected use of phones/tablets?

6. Experiment 1 procedure. What was the presentation level of the sounds and what device was used for presentation?

7. One final and very minor comment: in all three experiments, rhythmic irregularity was created by varying the position of the second, third or fourth word (W2, W3, W4) in a five-word sequence. However, it seems to me that varying the position of the second word is effectively the same as varying the position of the third word: if listeners assume the first IOI (i.e. the gap between W1 and W2) to be the “true” IOI, then it is the onset of W3 in these “W2-shift” stimuli which creates the percept of irregularity. The upshot of this is that there are effectively twice as many stimuli where attending to W3 (as opposed to W4) could enhance irregularity detection. A related, but slightly different, point is that stimuli where the irregularity occurs on W3 or W4 entail a phase (but not period) reset, whereas stimuli where the irregularity occurs on W2 could be thought of as entailing both a phase and period reset. Of course, both of these things would be the case for all presented conditions, and therefore shouldn’t impact between-condition comparisons – plus this comment only holds true if there’s no gradual build-up of temporal expectations throughout the course of the experiment (see my first point above). Basically, I’m just curious to know if the authors have thought about this at all!

### MINOR COMMENTS, TYPOS ETC. ###

p13, line 147: participant’s => participants’

p17, line 260: control of Type II => control Type II

p24, lines 427-434: check direction of quotation marks in parenthesised scales

p24, line 435: participants’ => participants

p28, line 524: performance the block => performance in the block

p29, line 564: those experiment => those experiments

p41, line 849: amplitude modulations peaks => amplitude modulation peaks

p43, line 905: perception can only => perception which can only

Many thanks to the authors for a stimulating and enjoyable paper!

Reviewer #2: Summary

The manuscript reports the results of three experiments conducted with spectrally degraded acoustic signals resembling real speech to different degrees. The goal of the experiments is to demonstrate that the perception of certain timing regularities in the presentation of isolated words is only accurate when the acoustic signals are interpretable as spoken language. I think the studies and their results may be publishable in some form, though as things stand, the current conclusions are not warranted by the data. Overall, the idea of speech rhythm and the experimental design are not particularly well informed by the literature. The experiments are based on a number of assumptions that lack direct support (especially the measurement of the regularity vs. irregularity condition), and the actual intelligibility of the stimuli is not adequately measured for all stimuli. I elaborate on the critical points below and offer some suggestions.

Major comments

(1) One major issue of the manuscript is the lack of an informed discussion and a clear working definition of the much-debated notion of speech rhythm, though it is in focus of the study. The introduction describes the authors’ own research and lacks a broader embedding in the relevant literature on the topic. There is an assumption that rhythm is necessarily isochronous (e.g. line 183 refers to “perfectly rhythmic speech”), though the generalisation from isolated words produced in time with a metronome to “speech rhythm” across the board is certainly a quantum leap of faith in terms of what “speech rhythm” might be. In line 752, there is a change of gears to “acoustic rhythms”. What is it, exactly?

(2) The original stimuli that served as the acoustic donors to the different types of acoustic degradation were produced in time with a metronome, leading to the assumption that the productions were indeed rhythmically regular (or rather, the timing of word onsets was isochronous). However, this was not checked. I am missing a test that would document the success of the intended production. In addition, there is a (not very well spelled out) assumption that speaking with a metronome leads to a tight alignment of p-centres of the stimuli with the metronome beats. This is likely to be the case, though I’d like to see this assumption substantiated by some kind of an acoustic measurement to demonstrate that in the regular condition, the acoustic pauses between the words were variable while the p-centre distances were kept relatively constant (and where exactly these p-centres were located). An overview of mean pause duration and mean inter-p-centre intervals (plus SDs or some measure of variability) is needed for the regular/irregular conditions. It might be helpful to get a linguist/phonetician on board, to help with sorting this out in an informed way. There is a substantial amount of literature out there (not referred to in the paper) showing that p-centre location falls close to the vowel onset, often coinciding with the moment of a fast spectral change. The authors claim acoustics do not play a role in the perception of speech rhythm (lines 800 – 806) but no acoustic measurements are actually taken.

It is quite possible that the best way forward for the manuscript is to discuss the results in terms of the perception of the p-centre location in degraded speech (instead of “speech rhythm”). Linguistic p-centres stir up acoustic isochrony, making the acoustics of language obey other principles than non-speech sounds and there is quite a bit of previous research (not reviewed in the paper) that shows this. I think the manuscript can make an important contribution to this research in that it shows how the p-centre effect disappears as speech signal gets unintelligible. For this framing to work, an experiment comparing natural vs. 16-channel noise vocoded productions would be needed though, to show that 16-channel noise vocoding behaves in the exact same way as natural speech (if it does). I am also missing the control condition – an acoustically regular version of the stimuli with equal duration of the pauses. It should be judged as regular in all stimuli that do not sounds like speech.

(3) I don’t see much value in Experiment-1 with regards to the overarching research goal of the manuscript. The intelligibility of the stimuli was not tested here, and I doubt that 1-channel vocoding sounded like speech at all. Looking at Figure 1, I expect the 1-channel vocoded manipulation to sound as some (slightly filtered) noise. In noise, there is no reason to expect a p-centre effect. Rather, one would expect an acoustic pause duration effect (i.e. sequencies with more equal durations of intervening acoustic silences would be judged as more “rhythmic” I assume they sounded just like noise – in this case, the conclusion that the authors are making is not warranted: It is not the isochrony that listeners stop being able to judge in these stimuli, it is the p-centre that is no longer a structuring unit of perception. In a way, the manuscript would be strengthened if Experiment-1 is removed and replaced by an experiment comparing natural and noise-vocoded productions.

(4) Overall, the intelligibility of the stimuli is an issue. It is lacking in Experiment-1. Experiment-2 states that some noise-vocoded and spectrally rotated stimuli were perceived as “sometimes intelligible” by the listeners – can the authors be sure that this means listeners understood what was said? I have worked with spectrally rotated stimuli and don’t quite believe this. Perhaps this rather means “sometimes sounding similar to speech” (in contrast to noise that cannot be interpreted as speech at all). As we know there can be substantial differences between self-perception and the actual behaviour, I think an additional test might be needed to check how many words can indeed be correctly recognised in the acoustic signals of different degradation levels. In Experiment-3, it is assumed that listeners can only learn to understand sinewave speech upon training or exposure, though we know some listeners recognise such sounds as speech without even much priming (e.g. Rosner et al. 2003, JSHLR).

(5) I notice that were quite a lot of participants excluded from the analyses, due to various reasons. Arguably, this is not the most methodologically rigorous way of doing stats. I wonder if a different approach would have some merit, e.g. an analysis based on non-aggregated data of regular/irregular responses using a glmer-model? This approach would have many advantages, e.g. individual differences in the understanding of sine-wave speech without much practice can be estimated in Experiment-3 (some people naturally perceive such sounds as speech without having been primed first) and the direction of timing change (delayed/sped-up presentation of target words) can be estimated in Experiment-2. This way, meeting a threshold criterion is no longer a valid reason for excluding participants’ data from the analyses. Additionally, the data can be modelled in a more sophisticated way, by including individually variable factors such as intelligibility and potentially acoustic factors of the stimuli.

Minor comments

Lines 474 – 476: check your reference here, it does not seem to be correct, neither of the two studies supports the claim made

Line 511: a reference is needed here

Line 515: something is wrong with the sentence structure here.

Lines 653-654: state the exact number of participants with chance-level d’ who were excluded from the analyses

Lines 844-901: pure speculation without any obvious relationship to the results

I suggest that the authors make (some of) their stimuli available as part of the submission (or in a repository like osf), for a better appreciation of the experimental designs and the conclusions.

6. PLOS authors have the option to publish the peer review history of their article (what does this mean?). If published, this will include your full peer review and any attached files.

Reviewer #1: No

Reviewer #2: No

---

## [Decision Letter · Decision Letter 1]

9 Nov 2022

PONE-D-22-14693R1Intelligibility improves perception of timing changes in speechPLOS ONE

Dear Dr. Zoefel,

Thank you for submitting your manuscript to PLOS ONE. After careful consideration, we feel that it has merit but does not fully meet PLOS ONE’s publication criteria as it currently stands. Therefore, we invite you to submit a revised version of the manuscript that addresses the points raised during the review process. Overall, both reviewers find the manuscript much improved. Reviewer 2 still has concerns about the lack of p-centre quantification potentially biasing the results. I must confess I do not entirely follow the logic: if the regularity manipulation failed, then that, to my mind, biases *against* finding any effect of intelligibility on regularity detection, because all conditions would be irregular and thus any effect of intelligibility on temporal accuracy should be similar across all conditions.However, I would like to give the authors a chance to edit and respond and if they feel they can further address the concerns in case I am misunderstanding. I hope that it is helpful that all stimuli are available, so that any reader who doubts the manipulation could also perform any analyses they wished on the stimuli.

We look forward to receiving your revised manuscript.

Kind regards,

Jessica Adrienne Grahn

Academic Editor

PLOS ONE

Journal Requirements:

Reviewers' comments:

Reviewer's Responses to Questions

**Comments to the Author**

1. If the authors have adequately addressed your comments raised in a previous round of review and you feel that this manuscript is now acceptable for publication, you may indicate that here to bypass the “Comments to the Author” section, enter your conflict of interest statement in the “Confidential to Editor” section, and submit your "Accept" recommendation.

Reviewer #1: (No Response)

Reviewer #2: (No Response)

2. Is the manuscript technically sound, and do the data support the conclusions?

Reviewer #1: Yes

Reviewer #2: Partly

3. Has the statistical analysis been performed appropriately and rigorously? 

Reviewer #1: Yes

Reviewer #2: Yes

4. Have the authors made all data underlying the findings in their manuscript fully available?

Reviewer #1: Yes

Reviewer #2: No

5. Is the manuscript presented in an intelligible fashion and written in standard English?

Reviewer #1: Yes

Reviewer #2: Yes

6. Review Comments to the Author

Reviewer #1: Many thanks to the authors for this thoughtful and thorough revision. They have addressed all of my questions, and I think the revisions in general have substantially improved the manuscript.

I have a few very minor comments, as follows:

Figure 1 caption (line 201): “Stimuli used in the Experiments 1 and 2” => “Stimuli used in Experiments 1 and 2”.

Lines 391 and 630: Thank you for including the data from the pseudo-audiogram procedure. However, I wonder if it might be better to call the cited figures a “threshold” rather than a “range”, since you’re only concerned with one end of the scale (i.e., the quietest sound detectable).

Line 770: “…only in the group that is trained relatively early.” For clarity, please could you specify which group you are referring to?

Lines 832-833: “…participants who perceive sine-wave speech as ‘speech’ (i.e. more intelligible)…” The parenthesised addition notwithstanding, I still feel as though this gives too much of an impression of a category shift from “non-speech” to “speech”. I’d therefore be inclined to soften the claim even further, perhaps describing the perception as more ‘speech-like’ (or something similar).

Line 833: “increased sensitivity to detect deviations” => “increased sensitivity to deviations”

Lines 895-896: “rhythmic patterns between language (e.g., stress-timed vs. syllable-timed [4])…” I appreciate the authors’ incorporation of a comment regarding cross-linguistic differences in speech rhythm. However, I feel I should point out that the “rhythm class” hypothesis they cite is now considered rather dated and controversial (see e.g., White et al, 2012; Nolan & Jeon, 2014). I think it’s perfectly fine to point to differences between languages, but I’d use a different reference if possible and avoid mentioning “stress-timed” and “syllable-timed” languages.

*White, L., Mattys, S. L., & Wiget, L. (2012). Language categorization by adults is based on sensitivity to durational cues, not rhythm class. Journal of Memory and Language, 66(4), 665-679.

*Nolan, F., & Jeon, H. S. (2014). Speech rhythm: a metaphor?. Philosophical Transactions of the Royal Society B: Biological Sciences, 369(1658), 20130396.

Once again, many thanks to the authors for this thought-provoking paper, which I very much enjoyed having the opportunity to read again.

Reviewer #2: The manuscript is much improved, and accompanying clarifications provided by the authors are also very helpful indeed. Re: comment (1) - The terms and the assumptions of the study are now better spelled out, thank you. Re: my comment (2) - I appreciate the reluctance of engaging with the idea of a p-centre in-depth. My issue remains unaddressed though. I am not misunderstanding something, the calculation of the d-prime measure that quantifies participants' perceptions relies on p-centres being regular (hit) or irregular (miss). This (ir)regularity is an assumption that is not documented anywhere in the stimuli. The participants might be perceiving the timing of the unintelligible stimuli extremely well, better than the timing of intelligible stimuli even, but because timing is a hit/miss preconception that is not quantified in any way, the authors are bound to arrive at the conclusion that intelligibility improves perception of timing regularities. What if the timing is irregular in all of the unintelligible stimuli (i.e. under both hit and miss assumptions)? Re: comment (3) - I understand the rationale. Re: comment (4) - the issue has been partially addressed (does 1-channel noise-vocoded "speech" sound as speech at all?). Re: comment (5) - it was a recommendation, not a major issue.

7. PLOS authors have the option to publish the peer review history of their article (what does this mean?). If published, this will include your full peer review and any attached files.

Reviewer #1: No

Reviewer #2: No

---

## [Editor Report · Decision Letter 2]

29 Nov 2022

Intelligibility improves perception of timing changes in speech

PONE-D-22-14693R2

Dear Dr. Zoefel,

We’re pleased to inform you that your manuscript has been judged scientifically suitable for publication and will be formally accepted for publication once it meets all outstanding technical requirements.

Kind regards,

Jessica Adrienne Grahn

Academic Editor

PLOS ONE
---

## [Editor Report · Acceptance letter]

2 Dec 2022

PONE-D-22-14693R2 

Intelligibility improves perception of timing changes in speech 

Dear Dr. Zoefel:

I'm pleased to inform you that your manuscript has been deemed suitable for publication in PLOS ONE. Congratulations! Your manuscript is now with our production department. 

Kind regards, 

on behalf of

Dr Jessica Adrienne Grahn 

Academic Editor

PLOS ONE